# The Usefulness of Resistant Maltodextrin and Chitosan Oligosaccharide in Management of Gut Leakage and Microbiota in Chronic Kidney Disease

**DOI:** 10.3390/nu15153363

**Published:** 2023-07-28

**Authors:** Weerapat Anegkamol, Panumas Kamkang, Sittiphong Hunthai, Maroot Kaewwongse, Mana Taweevisit, Natthaya Chuaypen, Pakkapon Rattanachaisit, Thasinas Dissayabutra

**Affiliations:** 1Metabolic Disease in Gastrointestinal and Urinary System Research Unit, Department of Biochemistry, Faculty of Medicine, Chulalongkorn University, Bangkok 10330, Thailand; 6174767730@student.chula.ac.th (W.A.); 6370041030@student.chula.ac.th (P.K.); 6370056530@student.chula.ac.th (S.H.); natthaya.c@chula.ac.th (N.C.); pakkapon.r@chula.ac.th (P.R.); 2Division of Physiology, School of Medical Sciences, University of Phayao, Phayao 56000, Thailand; maroot.ka@up.ac.th; 3Department of Pathology, Faculty of Medicine, Chulalongkorn University, Bangkok 10330, Thailand; dr.mana4@gmail.com; 4Center of Excellence in Hepatitis and Liver Cancer, Department of Biochemistry, Faculty of Medicine, Chulalongkorn University, Bangkok 10330, Thailand; 5Department of Physiology, Faculty of Medicine, Chulalongkorn University, Bangkok 10330, Thailand

**Keywords:** gut microbiota, prebiotics, chronic kidney disease, chitosan oligosaccharide, maltodextrin, gut leakage, tight junction

## Abstract

Microbiota-dysbiosis-induced gut leakage is a pathophysiologic change in chronic kidney disease (CKD), leading to the production of several uremic toxins and their absorption into the bloodstream to worsen the renal complications. We evaluate the benefits of resistant maltodextrin (RMD) and chitosan oligosaccharide (COS) supplements in cell culture and CKD-induced rats. The RMD exerted a significant anti-inflammatory effect in vitro and intestinal occludin and zonula occluden-1 up-regulation in CKD rats compared with inulin and COS. While all prebiotics slightly improved gut dysbiosis, RMD remarkably promoted the relative abundance and the combined abundance of *Lactobacillus*, *Bifidobacteria*, *Akkermansia*, and *Roseburia* in CKD rats. Supplements of RMD should be advantageous in the treatment of gut leakage and microbiota dysbiosis in CKD.

## 1. Introduction

Chronic kidney disease (CKD) is a global health crisis. Patients with CKD suffer from the retention of serum uremic toxins that subsequently instigate systemic complications. Uremic toxins are mainly produced in the intestine by the gut microbiota and absorbed into the bloodstream. In CKD patients, harmful gut microbiota overgrowth leads to the production of uremic toxins, inflammation, and an increase in gut permeability [1,2]. The retention of serum uremic toxins such as urea, ammonia, indoxyl sulfate, P-cresol, and trimethylamine-N-oxide causes fatal cardiovascular and metabolic complications and subsequently worsens the renal function.

Several approaches to suppress uremic toxin synthesis and absorption have been recently introduced to alleviate CKD progression or induce renal recovery. Among these, dietary supplementation with prebiotics and probiotics was one of the most promising methods [3,4,5,6]. Prebiotics serve several beneficial functions in gut ecology. They act as a food source to promote the growth of commensal gut microbiota. Additionally, they are fermented and metabolized into short-chain fatty acids (SCFA), which provide more favorable functions, including anti-inflammation, suppression of pathogenic bacteria, immunomodulation, mucin production, maintenance of a normal gut barrier, and increased colonic mineral absorption [7]. Inulin, or fructo-oligosaccharide (FOS), and galacto-oligosaccharide (GOS) are common prebiotics used in various products and were evidently beneficial in correcting gut dysbiosis and lowering serum uremic toxins [8,9,10].

Recent studies disclosed the usefulness of other prebiotics in modifying gut microbiota and modulating gut and other systemic diseases. Chitosan, as a non-plant derived prebiotic, was used to treat bowel inflammation and metabolic syndrome [11,12], and maltodextrin, a starch commonly used as a placebo, was lately reported to contain prebiotic properties [13]. In the present study, we focused on the effects of chitosan oligosaccharide (COS) and resistant maltodextrin (RMD) supplementation on the gut barrier integrity in cell culture and cisplatin-induced CKD rats. We anticipated that COS and RMD supplementation may improve the gut barrier and alleviate gut dysbiosis found in CKD.

## 2. Materials and Methods

### 2.1. Tested Chemicals

The resistant maltodextrin (RMD) and inulin used in this study were purchased from Banpong Novitat Co., Ltd. (Bangkok, Thailand). RMD is an oligosaccharide with random 1–2, 1–3, 1–4, and 1–6 α and β glycosidic linkages, making it partially resistant to human digestive enzymes. The glycemic index of RMD was evaluated to be approximately 59, and it has a 2000 Da molecular weight. Inulin is a chain of fructose molecules linked together, with a glucose molecule at one end. The molecular weight of inulin typically ranges from approximately 500 to 4000 daltons.

Chitosan oligosaccharide (COS) is an oligomer composed of β-(1-4)-linked d-glucosamine units. It is derived from the deacetylation and hydrolysis of chitin. The molecular weight of COS is less than 10,000 daltons, and each molecule consists of approximately 55 monomers. A small molecule COS used in this study was synthesized and provided by Assist. Prof. Dr. Rath Pitchyangura from the Department of Biochemistry, Faculty of Science, Chulalongkorn University, with a molecular weight of around 5–9 kDa.

### 2.2. In Vitro Study

#### 2.2.1. MTT Survivability Test

Caco-2, human colon adenocarcinoma cells were used in this study. The cells were cultures in sterilized DMEM (Dulbecco’s modified Eagle’s medium, containing 4 mM L-glutamine, 4500 mg/L glucose, 1 mM sodium pyruvate, and 1500 mg/L sodium bicarbonate) supplemented with 10% *v*/*v* FBS (fetal bovine serum), 1% *v*/*v* penicillin, and 1% *v*/*v* streptomycin and incubated at 5% CO_2_ and 37 °C. The cells were seeded in a 96-well plate with 15,000 cells per plate and incubated for 24 h. The cells were washed with PBS (phosphate-buffered saline) and new DMEM was added to the test substances for 24 h. The test substances included inulin, RMD, and COS at 10, 50, 100, 500, and 1000 µg/mL. PBS, 0.05% acetic acid, and 0.3% H_2_O_2_ were used as positive control, vehicle, and negative control for viable cells, respectively. After 24 h, 3-(4,5-dimethylthiazol-2-yl)-2,5-diphenyltetrazolium bromide (MTT) was added to the cells. A multidetection microplate reader was used to detect the absorbance at 570 nm on the culture plate. The survivability of cells exposed to different concentrations of the test substances were compared to the positive control.

#### 2.2.2. Transepithelial Electrical Resistance (TEER) Assay

Caco-2 cells of 500,000 cells were seeded into transwell (ThinCertTM, 0.4 μm pore size, 1.131 cm^2^ culture surface) and incubated for 21 days with high glucose DMEM supplemented with 10% FBS, 1% penicillin, and 1% streptomycin. Then, the cells were washed by PBS and the medium was replaced by FBS-free media. The cells’ TEER values were measured using an epithelial voltohmmeter (EVOM2) to obtain TEER values at 0 h. The cells in the transwell were treated with PBS, as a control, and the tested substances, including 100 µg/mL of inulin, COS, and RMD. A concentration of 60 ng/mL of TGF-β, growth factor, was used to see the TEER value in integrity-enhanced epithelial tissue. The study was carried out in 2 settings: normal and inflammation. In the inflammation setting, the cells were challenged by adding 50 ng/mL TNF-α, pro-inflammatory cytokine, in both the apical and basolateral sides to induce cell inflammation that disturbs the epithelial integrity of the cells 1 h before adding the test substances. The cells from each treatment were incubated for 24 h and then their TEER values were remeasured to obtain TEER values at 24 h. Fold changes were calculated from the ratio between TEER values at 24 h and TEER values at 0 h. The fold change in TEER values from each treatment was compared to the fold change in TEER from the control to calculate the %TEER to control.

### 2.3. Animal Study

#### 2.3.1. Animal Preparation, CKD Induction, and Sample Collection

The rats were raised in the Animal Center at the Faculty of Medicine, Chulalongkorn University, in a 35 cm × 75 cm × 18 cm cage, at 22 °C, under a 12/12 light and dark cycle, 40–60% relative humidity, with ad libitum food and drink. Four-week-old Wistar rats were purchased from Nomura Siam Co., Ltd. and acclimatized for two weeks before the initiation of the experiment. The rats were randomly assigned to five groups: control, CKD, inulin, COS, and RMD, each consisting of 6 animals. To induce kidney damage, the rats in the CKD, inulin, COS, and RMD groups received an intraperitoneal injection of cisplatin at a dosage of 10 mg/kg of body weight. In contrast, the control group rats were injected with 1 mL of normal saline. Following the injection, all rats were monitored for a period of five weeks to allow the acute renal injury to subside.

During this period, the rats were provided with normal chow (CP 802), comprised of carbohydrate 52%, fat 19.77%, and protein 28.23%, and drinking water containing 1% *w*/*w* phosphate (as phosphoric acid), along with the designated treatments for the experiment. Additionally, the rats were orally administered 54 mg of inulin per kg, 16 mg of COS per kg, or 54 mg of RMD per kg of rat. An approximately 0.5 mL blood sample was obtained from each rat’s tail vein using heparinized 1.5 mL microcentrifuge tubes. The blood was then centrifuged at 2500 G for 10 min to separate the plasma, which was subsequently stored at −80 °C. The rats’ fresh feces were collected and preserved in DNA/RNA Shield^®®^ reagent (ZYMO Research, Irvine, CA, USA) at −80 °C.

Blood samples were collected from all rats before the administration of cisplatin, at the start of the treatments, and on the day of sacrifice. At the end of the twelve-week study period, the rats were euthanized using long exposure to CO_2_. On the day of sacrifice, blood samples were collected via cardiac puncture. The kidneys were longitudinally split, with half of the specimens placed in 4% paraformaldehyde and stored at 4 °C, while the other half was collected in RNAlater^TM^ and stored at −80 °C. The jejunum section of the intestine was cut into 1 cm pieces and stored at −80 °C. Fecal samples were also collected and stored in DNA/RNA Shield^®®^ reagent (ZYMO Research) at −80 °C. The remaining sections of the intestine and right femur bones were collected and preserved in paraformaldehyde at 4 °C.

#### 2.3.2. Gene Expression of Tight Junction Protein

RNA was extracted from the jejunum sections of the intestinal tissue, and quantitative reverse transcription-polymerase chain reaction (RT-qPCR) was performed. RT-qPCR utilized primers specific to GAPDH (a housekeeping gene), claudin-1, occludin, and zonula occludens-1 (ZO-1). The goal was to measure the relative expression of each tight junction protein in response to different treatments.

To determine the fold changes of each tight junction protein relative to the control group, the expression levels were calculated and normalized to the housekeeping gene. This analysis allowed for an assessment of the impact of each treatment on the expression of these crucial tight junction proteins.

#### 2.3.3. Serum Creatinine, Calcium, Phosphate, and PTH Profiling

The concentrations of creatinine, calcium, and phosphate in the samples were analyzed using the automated Alinity ci system at the Department of Laboratory Medicine, Faculty of Medicine, Chulalongkorn University. The serum parathyroid hormone (PTH) concentration from each treatment was determined using an enzyme-linked immunosorbent assay (ELISA) of a parathyroid hormone ELISA kit in 96-well plates purchased from Wuhan Fine Biotech Co., Ltd. (Wuhan, China). The optical densities were measured at a wavelength of 450 nm using multidetection microplate readers. In addition, the estimated creatinine clearance (eClCr) (Appendix A) was calculated using the web-based ACLARA (https://idal.uv.es/aclara/, accessed on 13 July 2023) [14].

#### 2.3.4. Histopathological Evaluation

The tissue samples were subjected to permanent slide-making processes at the Department of Pathology, Faculty of Medicine, Chulalongkorn University. Sections of jejunum tissues were stained using the hematoxylin and eosin (H&E) staining technique to visualize the ZO-1 protein. Immunofluorescent staining was performed using a rabbit polyclonal antibody specific to ZO-1. The evaluation of the stained tissues was conducted by a pathologist affiliated with the Department of Pathology, Faculty of Medicine, Chulalongkorn University.

#### 2.3.5. Intestinal Microbiota Analysis

The DNA extraction process from fecal samples involved using the ZymoBIOMICS™ DNA Miniprep Kit according to the manufacturer’s instructions provided by Zymo Research Corp and supplied by S.M.Chemical Supplies Co., Ltd., Bangkok, Thailand. The resulting DNA was then assessed for concentration and purity using a DeNovix™ UV-vis spectrophotometer (Purchased from Bio-Active Co., Ltd., Bangkok, Thailand) and stored at −20 °C until further analysis.

To analyze the intestinal microbiome, the 16S/ITS Microbiome Profiling Service offered by Modgut Genomic Service at King Mongkut’s University of Technology Thonburi (Thailand), was utilized. The V3–V4 region of the 16S rRNA gene, serving as the target sequence, was amplified. The richness of microbial taxonomic groups in the samples was determined by the number of groups, while the evenness of their distribution indicated the evenness of the groups. For data visualization, an analysis of the alpha diversity, which includes the Shannon index and Pielou’s evenness, was conducted to summarize the ecological communities of the gut microbiota in terms of richness and evenness. The top relative abundance taxa were determined by calculating the average abundance of each taxon in each group divided by the total abundance of that taxon. The relative abundance of the bacteria was further analyzed and correlated with physiological parameters.

### 2.4. Statistical Analysis

The statistical analysis in the current study was conducted using SPSS version 22.0. Continuous data were assessed using the Student *t*-test for comparing two independent groups, while the ANOVA with post hoc Bonferroni test was used for comparisons involving more than two groups. Non-parametric variables were analyzed using the Mann–Whitney U test. In an animal experiment, the Student *t*-test was employed to compare two independent groups, and the Kruskal–Wallis test was used to evaluate differences in means for non-parametric data. To examine correlations, Pearson correlation was employed. Significance was determined at *p* < 0.05. Figures, diagrams, and graphs were created using GraphPad Prism 9 (Windows 64-bit) v9.5.1.733.

### 2.5. Ethical Consideration

The research adhered to the principles outlined in the Helsinki Declaration and followed the guidelines of good clinical practice when involving human participants, who provided fecal samples. The experimental procedures involving animals were conducted in accordance with the protocols of the Institutional Animal Care and Use Committee (IACUC). Ethical approval for the study was obtained from the Institutional Review Board of the Faculty of Medicine, Chulalongkorn University (IRB number 914/64). The protocols for animal experiments were approved by the Chulalongkorn University Animal Care and Use Committee (CU-ACUC protocol number 004/2563).

## 3. Results

### 3.1. In Vitro Study

In this study, we conducted an MTT survivability test on Caco-2 cells to assess the potential cytotoxicity of inulin, chitosan oligosaccharide (COS), and resistant maltodextrin (RMD). Our results demonstrated a significant decrease in cell viability at a concentration of 1000 µg/mL for inulin and COS, and at 500 µg/mL for RMD (Figure 1A). These findings indicated the presence of a toxic dose of the tested treatments on Caco-2 cells. Based on these results, we selected the concentration of 100 µg/mL for inulin, COS, and RMD to investigate the impact on transepithelial electrical resistance (TEER) across the Caco-2 cell monolayer.

To further investigate the effects of prebiotic treatments on intestinal barrier integrity, we employed the TEER assay. The TEER values of the Caco-2 cell monolayers were compared between each prebiotic treatment and the control group. As a positive control for TEER enhancement, TGF-β was included. In the normal setting, none of the prebiotic treatments exhibited a significant increase in the %TEER compared to the control group, as shown in Figure 1B. However, under an inflammation-induced setting using TNF-α induction, both inulin and maltodextrin demonstrated a substantial increase in TEER compared to the vehicle-treated group, as shown in Figure 1C.

### 3.2. Animal Studies

All rats with CKD exhibited significantly higher serum creatinine levels compared to the control group, indicating the induction of CKD through cisplatin administration. The estimated creatinine clearance of CKD rats was about 41.1% lower than the control (3.80 ± 0.42 vs. 2.24 ± 0.97 mL/min in control and CKD groups, respectively). However, no significant differences were observed between prebiotic-treated and untreated rats in terms of serum creatinine levels (Figure 2A), suggesting that the prebiotic treatments did not have a significant impact on renal function.

The RNA expression levels of occludin (OCLN) and zonula occluden-1 (ZO-1) in the jejunum of rats were analyzed. We found no significant differences in the RNA expression of OCLN among any of the groups. However, treatment with RMD resulted in a promotion of jejunal ZO-1 expression in rats with CKD, as shown in Figure 2B. This suggests that RMD may play a role in enhancing the expression of ZO-1, a critical tight junction protein involved in gut barrier integrity.

Immunohistochemistry studies also revealed that COS and RMD slightly rescued ZO-1 protein expression in the jejunum of rats with CKD, further supporting the potential protective effects of RMD on gut barrier integrity (Figure 2C).

### 3.3. Gut Microbiome Study

#### 3.3.1. Fecal Microbiota Diversity

The Shannon index and Pielou’s evenness were evaluated to assess the diversity and evenness of bacterial communities at week 12 of the experiment. However, no significant differences were observed in the Shannon index and Pielou’s evenness among the different groups (Figure 3A).

In contrast, when examining the control group, beta diversity analysis to compare the gut microbiota alteration between pre- and post-study in each treatment group using principal coordinate analysis (PCoA) of Bray–Curtis dissimilarity demonstrated no significant difference in diversity between week 0 and week 12. This suggests that the overall composition of bacterial communities in the control group remained relatively stable over the course of the experiment. The CKD group exhibited a significantly lower diversity at the 12th week, while inulin supplementation led to a relatively lesser but still significantly lower diversity. This indicates a notable change in the bacterial community composition over time in these groups. On the other hand, the rats treated with RMD did not show a significant change in beta diversity, similar to the control group, suggesting that RMD treatment maintained a relatively stable and consistent bacterial community composition throughout the 12-week period. Interestingly, the COS group displayed a significant increase in diversity, suggesting a notable shift in the composition of bacterial communities caused by the COS treatment.

#### 3.3.2. Relative Abundances and Correlation

Regarding the relative abundance of bacterial phyla in the fecal microbiota, comparisons were made between each treatment group and the CKD group (Figure 4A). At week 0, in the control group, *Cyanobacteria* were found to be higher in abundance compared to the CKD group, while *Proteobacteria* were lower. By week 12, significant differences in relative abundance emerged. In the control group, *Verrucomicrobiota* and *Campylobacterota* were higher in abundance compared to the CKD group. Similarly, in the RMD group, a higher relative abundance of *Verrucomicrobiota* compared to CKD was observed. These findings indicate that both the control and RMD groups exhibited higher levels of *Verrucomicrobiota* compared to the CKD group at the end of the 12-week period. These results indicated the potential influence of RMD on the relative abundance of the specific bacterial phylum *Verrucomicrobiota*.

The analysis of relative abundances at the genus level revealed slight differences in several bacterial genera across each treatment group; however, no significant differences were observed. While the relative abundances of various genera showed some variability, these differences did not reach statistical significance. Of particular interest were the beneficial bacteria, including *Lactobacillus*, *Bifidobacterium*, *Roseburia*, and *Akkermansia*. The relative abundances of these genera were aggregated for each treatment group. Remarkably, a significant increase was observed in the RMD treatment group compared to the other groups, as shown in Figure 4B. This finding suggests that the administration of RMD had a notable impact on the relative abundances of these beneficial bacterial genera. The significant increase in the relative abundances of the aforementioned genera in the RMD treatment group indicates the potential of RMD in promoting the growth and proliferation of these beneficial bacteria. These findings support the notion that RMD may exert beneficial effects on the gut microbiota composition by selectively enriching beneficial bacterial populations.

To explore the relationships between the fecal microbiota and various physiological parameters, including serum creatinine levels, ZO-1 RNA expression, and occludin RNA expression, Pearson correlation coefficients (*r*-values) were calculated. These correlation coefficients provide insights into the strength and direction of associations between variables. A heatmap was generated based on the correlation coefficients to visualize the relationships between the variables. The heatmap allows for a comprehensive overview of the correlation patterns among the measured parameters (Figure 4C).

## 4. Discussion

Prebiotic is a term used for a group of nutrients that are degraded and function as a food source by gut microbiota [15]. Several types of prebiotics are currently available. The most common prebiotics used in food products are fructans or inulin and fructo-oligosaccharide (FOS), which evidently promoted lactic acid bacteria growth, and galacto-oligosaccharide (GOS) that boosted *Bifidobacteria* and *Lactobacilli* [16]. Other prebiotics such as lactulose, arabinoxylans, xylo-oligosacharide (XOS), resistant starch, and other carbohydrate oligosaccharides are sometimes used. Currently, most scientists believe that prebiotics only work as the bacterial food source to maintain a balanced gut microbiota and restore gut microbiota homeostasis or gut eubiosis [17].

The present study focused on utilizing prebiotics to alleviate gut leakage. As an initial step, we tested candidate prebiotics to assess their ability to enhance tight junction protein expression and increase transepithelial electrical resistance, particularly in an inflammatory state, using in vitro models. Overall, our investigation demonstrated that in vitro studies could partially reflect the in vivo results. Prebiotics that showed stronger effects on Caco-2 cells tended to exert favorable effects in animal studies.

We used inulin as a standard prebiotic and found that even though inulin could restore transepithelial electrical resistance of inflammation-induced Caco-2 cells, in the in vivo study, inulin was ineffective in rescuing tight junction expression in the intestine. Lately, several groups of researchers utilized inulin as a prebiotic, or a component in synbiotics, to treat renal injury, and most of these studies found that prebiotics and/or synbiotics containing inulin improved serum creatinine, inflammatory cytokines, uremic toxins, glucose, and lipid profiles, but did not mitigate the pathological change in affected tissues, such as cardiac, kidney, or intestinal integrity [18,19,20]. Melekoglu, et al. reported that inulin supplementation to CKD rats could reduce serum creatinine, p-cresyl sulfate, and IL-6 but had no effect on serum indoxyl sulfate and colonic claudin-1 and occludin protein expression [21]. Regarding this, inulin is not the most appropriate prebiotic to treat kidney disease patients, and new prebiotics should be investigated. In the present study, we explored COS due to the previous report of microbiota-independent prevention of intestinal epithelial inflammation in vitro [22], and RMD, which has been commonly used as a placebo in numerous prebiotics studies, but also presents some prebiotic properties.

Chitosan is a polymer of randomly distributed acetylated and deacetylated forms of D-glucosamine which is derived from chitin found in the outer skeleton of shellfish. Chitosan has a lot of benefits as a biomedical polymer for tissue engineering, artificial organ synthesis, and wound healing [23]. Intake of chitosan may reduce intestinal fat absorption and was assumed to be advantageous in the treatment of hypertension, hypercholesterolemia, and obesity. The degradation of chitosan yields COS, which has lower viscosity and high intestinal epithelial absorption [24]. COS contains an anti-bacterial activity against certain microorganisms such as *Escherichia coli*, *Pseudomonas aeruginosa*, *Acinetobacter baumannii*, *Staphylococcus aureus*, etc. [25,26].

In the aspect of prebiotics, COS promotes Bifidobacteria, Lactobacilli, Prevotella, Rosuburia, and Faecalibacterium prausnitzii growth, while suppressing the number of Firmicutes, Streptococci, Bacteroides fragilis, Clostridium spp., and E. coli [27,28]. In certain conditions, COS improves gut dysbiosis and intestinal epithelial leakage [12]. Additionally, COS was reported to be beneficial in metabolic syndrome, diabetes mellitus and fatty liver disease since it can inhibit hepatic fat accumulation, reduce adipogenesis, and stimulate white fat cell browning and promote glucose homeostasis in diabetic rats [11,29,30,31]. Recent studies revealed the beneficial effects of chitosan nanoparticles and COS in ulcerative colitis, autoimmune encephalitis, autoimmune arthritis, lupus nephritis, and autoimmune hepatitis [32,33,34]; however, there have been a limited number of studies about COS in renal diseases. Chitosan nanoparticle-encapsulated drugs combined with metformin reduced creatinine, proteinuria, and downregulated TNF-α, IL-6, and IL-1β in type 2 DM rats [35,36]. Zhang H., et al., revealed that COS ameliorated proteinuria and the expression of kidney injury markers and may reverse pathologic change in diabetic rats’ kidneys [37]. Recent research revealed that COS attenuates oxidative damage, renal fibrosis, and renal cyst growth [38,39]. Accordingly, we expected that supplementation with COS could be beneficial in chronic renal disease. However, our in vitro study showed that COS could not enhance the integrity of colonic epithelial cells in normal or inflammatory states and giving COS supplements to CKD rats could not restore the intestinal tight junction protein expression and the glomerular function. The gut microbiome study demonstrated that COS slightly improved the number of Firmicutes and Patescibacteria, while reducing Bacteroidota, Vercurromicrobiota, and Desulfobacterota, similar to inulin feeding, and supposedly partially normalized gut dysbiosis. We infer that COS may not be an ideal prebiotic to treat gut leakage in CKD.

It should be noted that a dosage of 8 mg/kg of COS was used in this experiment, while 20 mg/kg of COS, given to animals in a previous study, was claimed to have an anti-inflammatory effect. Regarding this, we supposed that a higher dose or alternative preparation of COS may be advantageous in CKD. Further investigation is required to validate this hypothesis.

RMD is a polymer of D-glucose derived from the enzymatical process of starch, especially from plants. RMD has been used as the placebo control in many studies focusing on prebiotic efficacy. However, indigestible, fermentable, resistant maltodextrin was proven to have prebiotic functions, as consumption of RMD increased gut microbiota richness, particularly the *Bifidobacteria* count, and short chain fatty acid (SCFA) production [40,41,42]. Recent reports revealed that when compared to FOS, RDM intake was more efficient in the production of acetate, butyrate, propionate, and total SCFA, as well as lower in trimethylamine synthesis [13,43]. However, evidence was lacking about the impact of RMD on disease prevention, including intestinal epithelium and kidney health.

Our study showed that RMD treatment efficiently increased monolayer Caco-2 cell integrity in normal and inflammatory-induced conditions and promoted intestinal ZO-1 expression in CKD rats. These effects are stronger than inulin at the same dosage. The RMD supplement substantially boosted the relative abundance of gut microbiota in CKD rats, and partly suppressed *Verrucomicrobiota* and ‘bad bacteria’ *Campyrobacterota* numbers. In addition, we found that the combination of ‘good bacteria’ *Lactobacillus*, *Bifidobacteria*, *Akkermansia*, and *Roseburia* increased significantly in RMD-fed rats compared to their CKD counterparts. This group of bacteria is responsible for SCFA synthesis and mucous membrane protection. These results indicated that RMD supplementation was better than dose-dependent inulin and COS in the modulation of gut dysbiosis.

Gut bacteria such as *Lactobacillus*, *Bifidobacteria*, *Akkermansia*, and *Roseburia* are commonly recognized as beneficial microbiota. These bacteria have been found to enhance metabolism, reduce inflammation, improve intestinal barrier function, and maintain microbiota homeostasis. Their higher abundance is associated with positive outcomes in the treatment of diabetes mellitus, obesity, and inflammatory bowel disease (IBD) [44,45,46]. In CKD patients, lower levels of *Lactobacillus*, *Bifidobacteria*, *Akkermansia*, and *Roseburia* have been strongly linked to poorer glomerular function, malnutrition, and fatal complications [10,46,47]. Therefore, promoting the growth of these beneficial bacteria through prebiotics such as RMD is considered advantageous for CKD therapy.

Based on the correlation study between the relative abundance of gut microbiota and serum creatinine or tight junction protein expression (Figure 4C), significant correlations were observed between specific components of the gut microbiota and serum creatinine levels. Notably, *Oscillibacter* and *Lachnoclostridium* showed a negative correlation, while *Sellimonas*, *Eubacterium nodatum*, and NK 4A214 exhibited a positive correlation with serum creatinine. Serum creatinine is commonly utilized as a marker of kidney function, particularly glomerular filtration rate. Normally, creatinine production is independent of gut microbiota activity, as it primarily originates from muscle metabolism and glomerular filtration. However, recent research has indicated that the gut microbiota can influence various aspects of human health, including metabolism and inflammation, which may indirectly impact kidney function [48]. It is important to note that correlation does not imply causation and may be linked to factors such as the animal’s activities, hydration levels, and underlying health conditions that can influence creatinine levels independently of the gut microbiota.

Moreover, we demonstrated the deleterious effects of *Alistipes* and *Anaerovorax* on the expression of intestinal tight junction protein expression. Previous studies reviewed the increase in *Alistipes* abundance in bowel inflammation related to the downregulation of intestinal claudin-1, ZO-1, and occludin expression [49,50], while *Anaerovorax* was associated with low tight junction protein expression in heat stress [51]. Regarding renal diseases, these bacteria were reported to be elevated in CKD patients [52,53]. We speculated that the abundance of *Alistipes* and *Anaerovorax* could reflect the severity of gut leakage and uremic toxin absorption in CKD patients.

## 5. Conclusions

We proposed the resistant type of maltodextrin as a desirable prebiotic supplement in CKD patients to improve gut integrity, lower uremic toxin absorption, and modulate beneficial gut microbiota growth. RMD is generally safe and has no contraindication for hypertension, diabetes mellitus, or renal insufficiency. We also expected that COS may be useful in the management of CKD, but the optimal preparation and dosage should be validated. Our future plan includes utilizing COS and RMD as components of synbiotics to alleviate complications of CKD in animal models.

## Figures and Tables

**Figure 1 nutrients-15-03363-f001:**
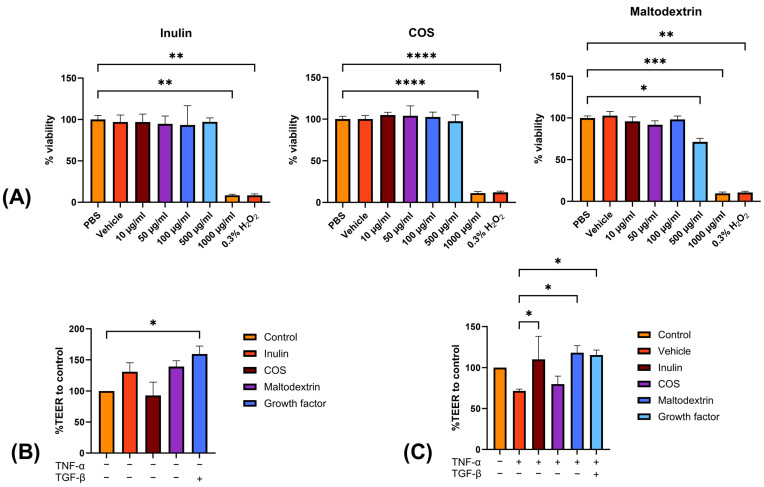
Effects of prebiotics on Caco-2 cells’ viability and TEER after 24 h of treatment. (**A**) MTT cell, (**B**) %TEER under normal conditions, (**C**) %TEER in an inflammation-induced condition. * *p* < 0.05, ** *p* < 0.005, *** *p* < 0.001, and **** *p* < 0.0001 as compared to PBS.

**Figure 2 nutrients-15-03363-f002:**
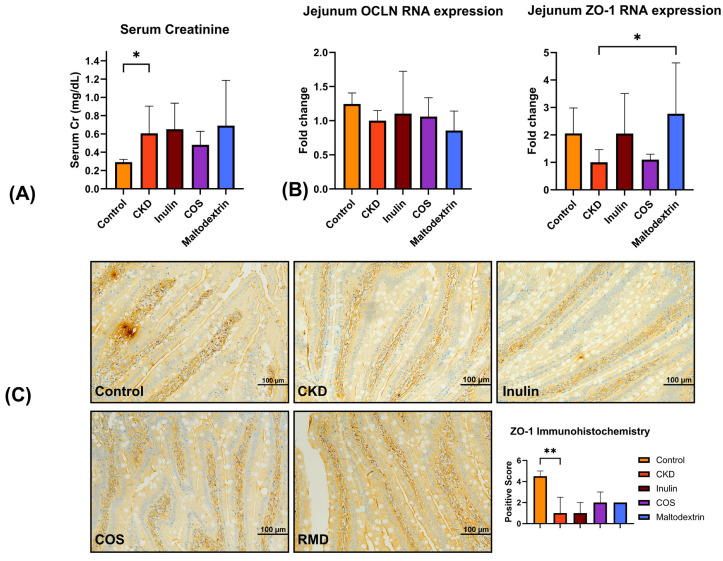
(**A**) Serum creatinine. (**B**) The RNA expression levels of occludin (OCLN) and zonula occluden-1 (ZO-1) in the jejunum. (**C**) Immunohistochemistry against ZO-1 protein in intestinal epithelium of rat jejunum. * *p* < 0.05, ** *p* < 0.005.

**Figure 3 nutrients-15-03363-f003:**
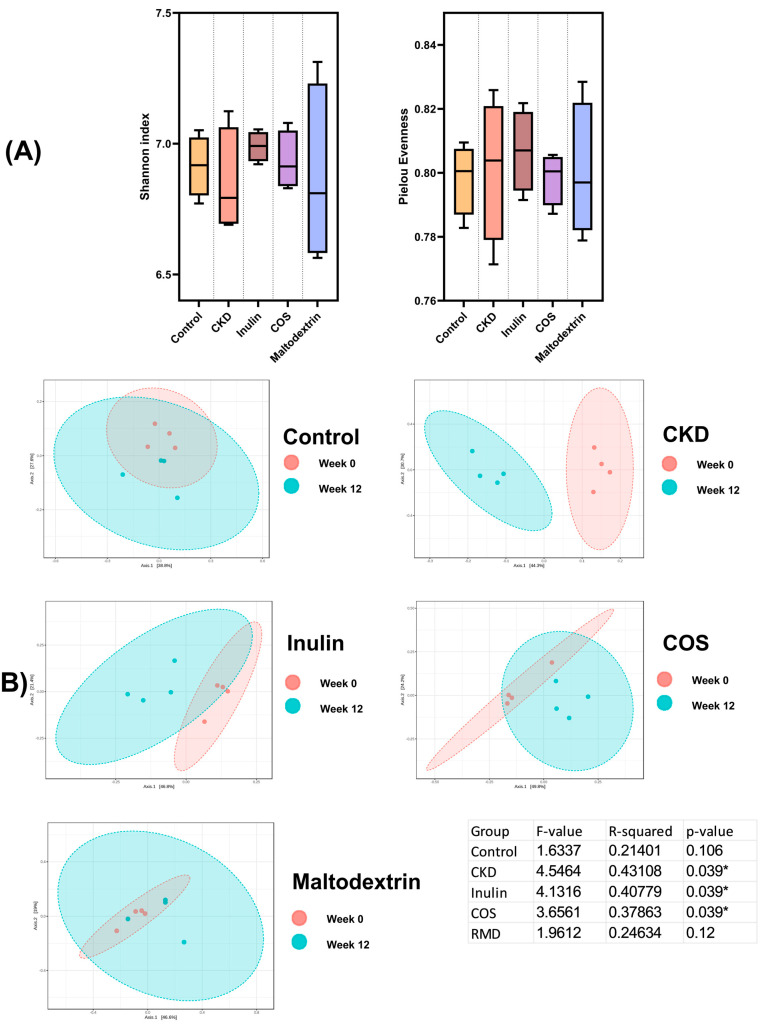
Species diversity of fecal microbiota: (**A**) alpha-diversity indices. The reported values are median ± IQR (*n* = 4). A significant difference among all the groups was not detected. (**B**) Beta diversity (Bray–Curtis dissimilarity) PCoA index. Results are means for *n* = 4; * *p* < 0.05.

**Figure 4 nutrients-15-03363-f004:**
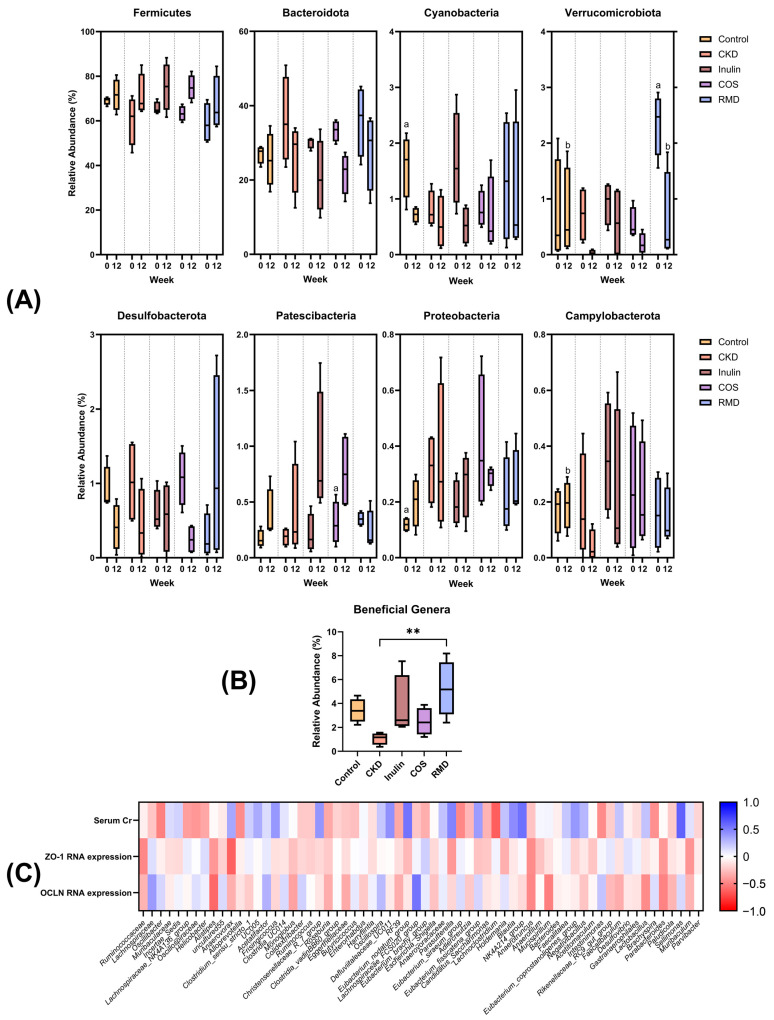
(**A**) Relative abundance of fecal microbiota at phylum level. (a) represents significant difference compared to CKD group at week 0, (b) represents significant difference compared to CKD at week 12. *p* < 0.05. (**B**) Relative abundance of 4 beneficial genera, *Lactobacillus*, *Bifidobacterium*, *Roseburia*, and *Akkermansia*, ** *p* <0.05. (**C**) Heatmap of correlation between fecal microbiota, serum creatinine, ZO-1 RNA expression, and occludin RNA expression. *r*-values were calculated using Pearson correlation test. Red color indicates negative correlation, blue color indicates positive correlation.

## Data Availability

The data presented in this study are openly available at https://doi.org/10.6084/m9.figshare.23618073 (accessed on 2 July 2023).

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
