# Peer review of "The Usefulness of Resistant Maltodextrin and Chitosan Oligosaccharide in Management of Gut Leakage and Microbiota in Chronic Kidney Disease"

_nutrients, 2023, doi:10.3390/nu15153363_

Round 1
Reviewer 1 Report
The authors have conducted interesting experimental research about the utility of resistant Maltodextrin and Chitosan Oligo-2 administration in CKD.
Minor points,
The Conclusions can be enriched with the authors' future perspective regarding this field.
Please insert de composition of the DMEM supplement in the text “4500 mg/L D-glucose, and L-glutamine”?
Line 33-35, after this phrase a ref. should be inserted, also after this sentence “ Inulin or fructo-oligosaccharide (FOS), and galacto…”
Attention to abbreviations and their meanings, once they are first inserted in the text no need for repetition afterward.
There are many grammatical errors in the manuscript that need correction, please revise and correct the whole manuscript. >Ex> Line 112, “RMD per kg of Approximately” , “of a parathyroid hormone” and so on.
Reviewer 2 Report
In this study, Weerapat Anegkamol and Colleagues, aimed to assess the effects of chitosan oligosaccharide (COS) and resistant maltodextrin (RMD) supplementation on the gut barrier integrity in cell culture and cisplatin-induced Chronic kidney disease (CKD) rats. They evaluated the benefits of resistant maltodextrin (RMD) and chitosan oligosaccharide (COS) supplement in cell culture and CKD-induced rats. They found that RMD exerted significant anti-inflammatory effect in vitro and intestinal occludin and zonula occluden-1 (ZO-1) up-regulation in CKD rats comparing with inulin and COS. Of interest, RMD remarkedly promoted the relative abundance and the combined abundance of Lactobacillus, Bifidobacteria, Akkermansia, and Roseburia in CKD rats. Therefore they concluded that supplement of COS and RMD could advantageous in the treatment of gut leakage and microbiota dysbiosis found in CKD.
The study is of interest since addressed the role of microbiota dysbiosis-induced gut leakage as a pathophysiologic factor in CKD. However, some points deserve further details and should be addressed.
-Animal Studies: the authors reported that "All rats with CKD exhibited significantly higher serum creatinine levels compared to the control group..". However, could they further specify whether there was a different degree of serum creatinine elevation?
-Relative Abundances and Correlation: Regarding animal preparation procedure, could authors described rat diet details? This is of relevance to evaluate differences in microbiota profile changes.
-Discussion: To improve the clinical impact of this study, the authors should recall and discuss the potential impact of Microbiota dysbiosis also in immune-mediated chronic diseases. The authors properly stated that "COS was reported to be beneficial in metabolic syndrome, diabetes mellitus and fatty liver disease.." . Very importantly, a link between intestinal microbiota, intestinal mucosal permeability and chronic immune-mediated diseases has been demonstrated. For instance, both chronic intestinal diseases such as celiac disease and autoimmune liver diseases significantly develop an immune response to Saccharomyces cerevisiae antigens, as result of over expressed intestinal mucosal immunoreactivity, as previously demonstrated (Anti-Saccharomyces cerevisiae and perinuclear anti-neutrophil cytoplasmic antibodies in coeliac disease before and after gluten-free diet. Aliment Pharmacol Ther. 2005 Apr 1;21(7):881-7. doi: 10.1111/j.1365 2036.2005.02417.x.; Anti-Saccharomyces cerevisiae antibodies (ASCA) and autoimmune liver diseases. Clin Exp Immunol. 2003 Jun;132(3):473-6. doi: 10.1046/j.1365-2249.2003.02166.x. ). The importance of gut microbiota profile and probiotic therapy in inflammatory/autoimmune diseases has been reported and supported in different conditions as recently suggested (Editorial: gut microbiota profile in patients with autoimmune hepatitis-a clue for adjunctive probiotic therapy? Aliment Pharmacol Ther. 2020 Jul;52(2):392-394. doi: 10.1111/apt.15795.).
Reviewer 3 Report
The manuscript presents a substantial amount of research work. It is interesting and provides insights into the effects of probiotics in the treatment of gut leakage and microbiota dysbiosis. However, the paper's description of the results is overly simplistic and confusing. Specific comments are as follows:
1. lines 39-41, references should be added.
2. The introduction section is too simplistic and should provide more detailed examples and principles about the effects of prebiotics on gut microbiota ecology.
3. In line 196, the author chose 500 µg/mL of inulin, COS, and RMD for further experiments. However, it was previously mentioned that 500 µg/mL of RMD was toxic. The author should provide an explanation for choosing this concentration and why it was not 100 µg/mL.
4. From line 246 to 248, the author suggests that the COS group exhibited significant changes compared to the control group and CKD group. However, based on Figure 3B, it can be observed that the COS group only showed significant changes in comparison to the control group, while no significant changes were observed when compared to the CKD group.
5. The author's description of the changes in gut microbiota is too simplistic, especially regarding Figure 4C. The author did not provide any analysis or description, and there is a need for a more in-depth analysis of the relationship between gut microbiota and physiological indicators.
6. The reference formatting is inconsistent, such as journal names, abbreviations, and page numbers. The author should adhere to the reference formatting requirements of Nutrients.
7. There is insufficient description of the Caco2 cells in the Results and Discussion section, making it difficult to comprehend the connection between the cell experiments and animal experiments.
Round 2
Reviewer 3 Report
The authors have addressed the concerns I had, and I agree that this manuscript could be accepted after mini revisions.